



**Title page:**
**A seismologist's beginnings: Inge Lehmann's experiences during the**
**1910s and '20s as a woman in science.**
**Lif Lund Jacobsen**
**Danish Nations Achieves, Kalvebod Brygge 34, 1560 Copenhagen V, llj@sa.dk**

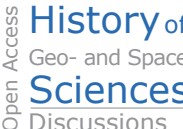

**Abstract**
Celebrated for her 1936 discovery of the Earth's inner core, seismologist Inge Lehmann (1888-
1993) has often been portrayed as a trailblazing female scientist, unwilling to accept discrimination
in her pursuit of an academic profession. Yet, a close reading of her experiences suggests that
Lehmann faced severe restrictions early on in her career. Only by being pragmatic about her
situation did she successfully establish herself as a professional scientist.
Having attended a progressive co-educational school before studying mathematics at the University
of Copenhagen, Lehmann had little direct experience of gender discrimination. After receiving her
bachelor's degree, she entered Cambridge University in 1911, along with Niels Bohr, but found
herself unprepared for the gendered social segregation practiced there. Exhausted from overwork,
Lehman abandoned her studies and returned to Denmark. Over the next six years, she came to
understand how severely her gender limited her career options.
In 1918, Inge Lehman returned to the University of Copenhagen to complete her studies, and
became a teaching assistant for a professor of actuarial science in 1923. Because her chances for
obtaining a scientific post at the university were slim, she joined Professor Niels Erik Nørlund in his
efforts to reform the Danish Geodetic Service. In 1928, Professor Nørlund rewarded Lehmann's
voluntary change of academic discipline from mathematics to seismology by appoint her as
Director of the Seismology Department.
[1]

## 1. Introduction

The Danish seismologist Inge Lehmann (1888-1993) is known for her 1936 discovery of the Earth's
inner core. Originally trained in mathematics, she began working as a seismologist in the mid-1920s
and continued in this field until the 1970s. For 24 years she headed the Seismology Department of
the Danish Geodetic Institute. When she began her work, it was rare for women to hold any
academic position. Yet, despite being the sole female in a male-dominated research community, she
soon gained international acclaim for her seismic research.
In many ways Lehmann's career is the story of personal success, of her scientific prowess
transcending her gender. Nevertheless, the road to success was difficult and full of challenges,
especially during her graduate and postgraduate years.  While she herself refused to accept any
notion of gender difference ascribed to her intellectual ability or interfering with her right to pursue
an (academic) career, society at large was less open-minded. In her early career, she felt this
discrimination keenly.



Using newly discovered unpublished historical documents, this article will document Inge
Lehmann's graduate and postgraduate years, examine to what degree her gender played a decisive
role in her experiences, and discuss the extent to which her experiences were representative of her
female contemporaries in academia.

***Table 1: Landmarks for women's rights in Denmark***
| 1875 Women are admitted to the universities (except to the study of theology). |
    |---|
| 1899 Married women achieve the same (financial) rights as unmarried women. |
| 1903 Girls are permitted to attend high school on equal terms with boys. |
| 1915 Women secure the right to vote. |
| 1919 Legislation passes regarding equal pay for equal work for civil servants. |
| 1921 Legislation passes that provides Equal Access for Women to All Public Service and Occupations (with the exception of clerical and military positions). |
| 1922 Married women gain the right to share in the legal custody of their children (but not have sole guardianship). |

Until recently, little was known about Inge's primary and secondary school years and her
mathematical studies at Copenhagen and Cambridge Universities (see, for example, Bolt and
Hjortenberg, 1994). New information about her life and career has been found in the documents and
correspondence that she bequeathed to her colleague, seismologist Erik Hjortenberg, who donated
them to the Danish National Archives in 2015. There, the Inge Lehmann archival collection consists
of twenty-one boxes of notes, letters, manuscripts and references. Additionally, a number of letters
from the 1910s and 1920s are held in the archival collections of Niels Bohr and Niels Erik Nørlund.
This material provides key insights into her early career. Letters from Inge and her family, recently
found by the author Lotte Kaa Andersen, provide a window on her childhood.

## 2. Childhood and schooldays
Inge Lehmann was the elder of two sisters who grew up in Copenhagen in an intellectual family.
Their mother, Ida ne Tørsleff (1866-1935), came from a family of booksellers. Several female
Tørsleff family members were part of the Women Rights Movement and significant public figures.
Inge's cousins served as head of the Danish Girl Scouts, chair of the Danish Women's Society, and
the Minister of Trade. Famously, her younger sister Signe, a single mother, became a school
superintendent.
Inge's father, Alfred Lehmann (1858-1921), had a Masters Degree in Applied Science from
Copenhagen Polytechnic. He established psychology as an independent research subject in
Denmark after he set up a private Psychophysics Laboratory for experimental psychological
research in 1886 (Moustgaard and Petersen, 1986). When the University of Copenhagen took over
the laboratory in 1890, Alfred Lehmann was appointed interim 'docent' (a teaching post ranked just
below professor). Financial constraints meant that had to take on additional paid work until 1911, as
the censor at a teachers' college, a librarian at the Royal Veterinary and Agricultural University, and
as a technical drawing teacher. Not until 1910 was he appointed 'ekstraordinær professor'
(professor without chair); and nine years later he was elevated to a professorship with chair. Alfred
Lehmann's substantial number of scholarly publications on experimental and applied topics range
from how emotions influence blood circulation, and the existence of occult phenomena (of which
he was skeptical), to studies of the maximum yield of physical and intellectual work (for detailed
descriptions of Alfred Lehmann's work, see Funch, 1986; and Pind, 2019).

Inge's parents had progressive views on education. In 1894 they enrolled her, and later her sister
Harriet, at Hanna Adlers Fællesskole, the first co-educational school in Copenhagen where girls and
boys were taught the same subjects together. This was highly unusual – most schools had separate
academic tracks for boys and girls. For intellectually inclined girls, gender-segregation policies
went even further. Exposing girls to intellectual exhaustion and stress during puberty was
considered harmful. Hence, girls under seventeen years old were prohibited from taking the high
school entrance exam, whereas boys, who were considered better suited biologically for such
activities, could take the exam and enter upper-secondary school (high school) at age fifteen
(Larsen, 2010). This policy persisted until 1903.
The experience of the founder of the school, Hanna Adler, as a woman in academia, inspired her to
establish her co-educational school.  In 1892, seventeen years after the University of Copenhagen
admitted its first women students, Adler (1859-1947) and Kirstine Meyer (1861-1941) were the first
two women to earn Master's Degrees in Physics. Meyer was also the first woman to gain a
habilitation in Physics, the traditional prerequisite of a professorship. Inspired by advanced
pedagogy in the USA, Adler opened her school a year after completing her degree. As teachers, she
hired several of her female co-graduates who were excluded from many of the jobs open to male
academics. At that time, women could not get university positions and, although their degrees
qualified them to teach at upper-secondary school (high school) level, many female college
graduates found work as primary school teachers. As a trailblazing female academic, Hanna Adler
firmly believed in gender equality. She was also the aunt of the physicist and Nobel laureate, Niels
Bohr, and a frequent guest in the Bohr household.
In autobiographical notes Inge Lehmann described her schooldays as happy, marked by serious
study without differential treatment of boys and girls (RA: Lehmann autobiographical note, [ca
1970]: W84-258078).[2] Inge showed considerable talent in mathematics and physics, and her father
was keen for her to pursue a degree in science. Kirstine Meyer taught her physics, and Thyra Eibe
(1886-1955), known for her expert translation of Euklids *Elementes,* taught mathematics. These
female scientists were uniquely qualified to support Inge Lehmann's academic ambitions. With such
role models, it is not surprising that the girl developed a strong sense of intellectual entitlement and
belief in gender equality.

*Figure 1: Inge Lehmann (to the right) with fellow High School graduates, 1906 – the first year*
*females were allow to graduate on equal terms (Anon (1918) Frk. H. Adlers Fællesskole 1893-*




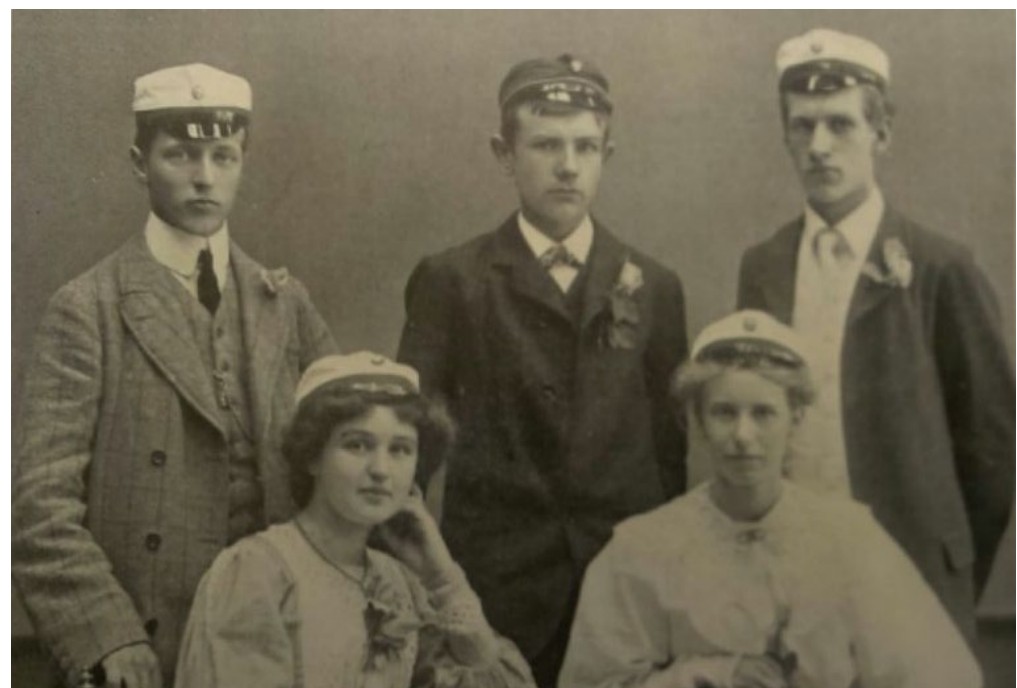


### 3. Studies at the Faculty of Science, University of Copenhagen

After passing her upper-secondary school graduation exams in 1906, Inge Lehmann worked as a
private tutor before, in the autumn of 1907, starting her studies at the Faculty of Sciences in the
University of Copenhagen – Denmark's only university, majoring in mathematics.


Between 1875 and 1925, 369 women sat for final examination at the University. Of that total, 326
did so after 1900, when the overall number of students had also increased from between 2,100-
2,300 at the turn of the century to approximately 4,500 enrolled in the university in 1925. In the
Faculty of Mathematical Sciences, the first precise student count dates from 1912, at which point
146 students were enrolled, 22 of them women (for details on early female students at Copenhagen
University, see Grane and Hørby, 1993; Rosenbech, 2014, Phil, 1983). Thus, when Inge Lehmann
started at the Faculty, female students were no longer a rarity, but neither were they numerous.

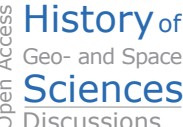


So far, no sources have been found that describe Lehmann's university experiences. She is not
mentioned in records linked with any other leadings students at the faculty, such as Niels Erik
Nørlund in mathematics or Niels Bohr in physics. Nor was she part of the interdisciplinary study
group, *Ekliptika*, which had a number of women participants (Pind, 2014). Lehman lived at home
and, evidently, focused entirely on her studies. The first part of her degree examination in the
summer of 1910 resulted in fine grades (RA: Københavns Universitet, Karakterprotokol Matematik,
[1908]: 2. del).[3]

**3.1 Studies at Newham College, Cambridge University**
After graduation, Inge Lehmann was eager to study abroad. In the spring of 1911 she entered
Newnham College, one of two women's colleges at Cambridge University, UK. Cambridge was
renowned for its excellence in mathematics. A form of examination unique to the university, the
Mathematical Tripos covered theoretical and applied mathematics, plus subjects in astronomy and
physics: it was notorious for its scope and difficulty. The exam was considered so challenging that
preparation traditionally involved equal parts theoretical study and physical activity – training both
body and mind in order to strengthen the intellect. Even after modification in 1909 to counter
falling enrollment and accommodate students' needs to specialize within one subject, the
Mathematical Tripos remained exceedingly demanding and equally prestigious (Warwick, 2003).
By choosing to read mathematics at Cambridge, Lehmann revealed the depth of her ambition, but
the English university's setting proved to be quite unlike what she had known in Copenhagen.
Women had been eligible to sit for the Tripos since 1881. Yet, although they could attend lectures,
women could not matriculate, attain full university membership, or be appointed to academic posts.
Only in 1948 were women admitted to Cambridge on equal terms with men. Un-matriculated
female students were denied access to laboratories and libraries. Since individual tutoring at
Cambridge often took place in conjunction with lab work, female students found themselves
prohibited from taking part in practical, hands-on experimentation, and could not be tutored by the
male lectures (for further details on the experiences of female academics at Cambridge University,
see, e.g., Evans, 2010; Richmond, 1997).
At Cambridge, the regular system of tutors, grants and student clubs was the prerogative of men,
and this further marginalized female students. During the 1880s and 1890s, therefore, a parallel



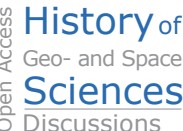

system of laboratories, libraries and tutors exclusively for female students had gradually built up
around the two women-only colleges, Girton and Newham.
While Inge Lehmann knew about similar parallel systems in Denmark – the Women's Reading
Society (Kvindelig Læseforening), for example – she had not experienced the degree of gender
segregation that prevailed in Cambridge. Even though the examination system at Cambridge was
reformed in 1909, and a number of vital resources were made available to female students via their
colleges, it was still difficult for women to study freely. In particular, restrictions imposed on
socializing between students of different sexes were far more onerous in Cambridge than in
Copenhagen, and they posed a real obstacle to the sharing of knowledge.
This was alien territory for Inge Lehmann, and her frustration about her experiences were expressed
in her correspondence with Niels Bohr, who was also coming to Cambridge.

*Figure 2: Newham College (Inge Lehmann Collection, The Danish National Archives)*

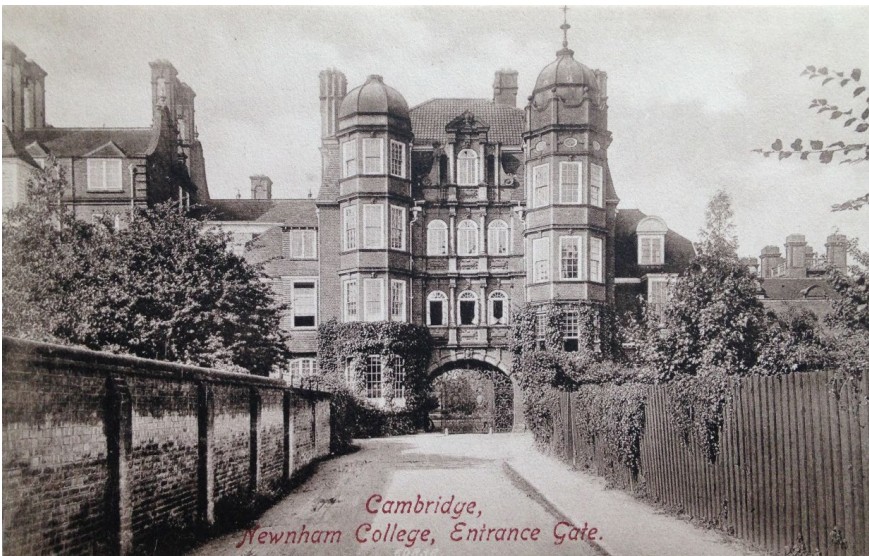



Niels Bohr completed his doctoral dissertation – *Studies on the Electron Theory of Metals* (*Studier*
*over Metallernes Elektronteori*) – in the spring of 1911 and planned to spend time at Cavendish
Laboratory in order to follow the experimental work of J. J. Thomson, the physicist.



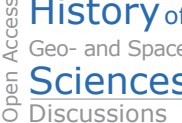

Bohr's interaction with Inge Lehmann in Cambridge is detailed by Aaserud and Heibron (2013). In
May 1911, he wrote his first letter, asking for her help to find out which physics lectures would be
relevant to his areas of interest, laid out in the enclosed a copy of his doctoral dissertation. After
reading the manuscript, Lehmann brief outlined the lectures she thought he might find useful and
ended her letter expressing hope that they could meet up when he came to Cambridge (NBA: I.
Lehmann letter, 2. Mai 1911).[4] This proved considerably harder than she had envisaged.
Bohr arrived in Cambridge at the end of September 1911. By early October, he had found an
apartment with the help of Lehmann and her network of friends. Over the next few months, Niels
Bohr and Inge Lehmann visited one another numerous times, but arranging these visits was always
troublesome: according to university regulations, Inge had to be chaperoned when spending time in
the company of a man.
On one occasion, shortly after Niels arrived in Cambridge, he was invited to Peile Hall, where Inge
Lehmann lived at Newnham College. Their meeting was only possible because Newnham's Vice-
Principal, Miss Strachey, had agreed beforehand to be present (NBA: Lehmann letter, n.d. [1911].[5]
Another visit had to be cancelled because Inge was unable to find a suitable chaperone on a Sunday
(NBA: I. Lehmann letter, 13. October 1911).[6]
A dinner party in early December 1911 proved particularly challenging. Inge was traveling to
Copenhagen to spend Christmas with her family, so Niels invited her, along with two male
mathematicians, to a farewell-dinner at his lodging. Before she could accept his invitation, Inge had
to ask him for the name of her chaperone. With that information, she could ask the principal of
Newnham Hall for permission to attend. She regretted the trouble, but wrote with resignation: "…
Cambridge is Cambridge" (NBA: I. Lehmann letter, 5. December 1911b).[7] Wise from experience,
Bohr had already arranged for a friend to attend the dinner party with his sister. Unfortunately,
Lehmann informed him, that sister was also a student at Newnham College, and her presence would
not fulfil the requirements of effective supervision (NBA: I. Lehmann letter, 5. December 1911a).[8]
Eventually, the list of dinner guests grew so long that Bohr was afraid there would be no room for
them in his small apartment, or so he ironically wrote to Margrethe Nørlund, his fiancée.
This correspondence illustrates how the restrictive social conventions at Cambridge obstructed
interactions between students of different genders – including the exchange of knowledge. Inge
Lehmann unquestionably felt the restrictions most acutely, but Niels Bohr also grumbled about the
University's strict code of conduct, which he found quite absurd. Although Bohr was likely

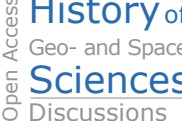

influenced by his free-thinking aunt, Hanna Adler, there can be no doubt that social conventions
between students of different sexes were far less cumbersome at the University of Copenhagen,
where no formalized system of gender segregation ever existed and teaching and practicums were in
effect co-educational.
Lehmann went home for Christmas in 1911, expecting to return to Cambridge for the start of the
spring semester. In March 1912, Bohr decided he had nothing more to gain from staying in
Cambridge and moved on to Professor Ernst Rutherford's laboratory in Manchester, where he spent
the next six months developing his pioneering atomic theory.
It was during Christmas break that Lehmann decided not to return to Cambridge for the next
semester. She had spent 1911 preparing for the Mathematical Tripos, and intended to sit for the
entrance exam in the spring of 1912. She was profoundly overworked. It has generally been
assumed that Lehmann abandoned her studies altogether because her recovery from this utter
exhaustion was so slow. She was literally unable to resume her university studies for a long time
(e.g. Bolt, 1997).
In reality, she was exhausted, but keen to return to Cambridge. Recently discovered correspondence
shows that Alfred Lehmann put a stop to her plans by refusing to fund her stay. Instead, he urged
her to seek employment in Denmark and make a living outside academia. In a letter to Inge written
in March 1912, her father explained his reasoning at length. Practically speaking, the rising cost of
living made it impossible for him to finance her studies any longer. Alfred's economic concerns
seem genuine, given his precarious employment at the University and his younger daughter
Harriet's recent enrollment at the Danish Royal Theatre's acting school. Yet Inge's health was of
primary importance. To protect his daughter, he could no longer in good conscience support
academic aspirations that were ruining her heath. To Alfred and many of his peers, it was a proven
fact that, whereas women might be as intellectual gifted as men, they lack the rigorous constitution
necessary for academic pursuits. College was better suited to the male disposition.
To argue his case Alfred Lehmann quoted several male professors of his acquaintance who strongly
believed that women did not have the mental stamina to meet the 'by no means unreasonable
requirements' for an MA in Copenhagen, let alone the more challenging studies in Cambridge. He
went on to relate "…a series of sad examples of how it went with intellectually gifted women who
wanted something more…". Their studies had made them so ill, they were forever in and out of



nerve clinics if not half insane. Not wanting the same fate for Inge, who already had shown signs of
fatigue, her father felt it would be irresponsible of him to let her continue with her studies. Instead,
he urged his daughter to seek practical clerical employment where she could undoubtedly rise to a
valuable and responsible administrative position in due time. Thus, there would be no need for her
to complete her final exam (Private: A. Lehmann letter, 11. March 1912).[9]
The biological argument that women were not equipped with enough energy and fortitude for
scientific studies likely originated in the rise of scientific medicine in the 19th century and, by
extension, the study of biological gender. From 1890 to the late 1910s, Doctor Leopold Meyer
published a series of influential medical texts in Denmark that problematized menstruation in
relation to physical and intellectual work: due to their female physiology, too much exertion of the
brain and nervous system would make women ill (Rosenbeck, 2014). Since Inge's father studied the
body's reaction to physical and intellectual work, he was most likely familiar with Meyer's ideas
and, therefore, concerned about his daughter's future in her chosen field.
Inge must have protested because Alfred – somewhat mollified – wrote again two weeks later to
suggest that she convalesce at home until September. Then, mindful of her health, she should
resume her studies at Copenhagen University. If her strength and her exam results were satisfactory
at the end of a year, he would find the necessary funds for another year at Cambridge, where she
could complete her MA-degree without sitting for the Mathematical Tripos. Ultimately, Alfred
thought it ill-advised for Inge to pursue a foreign degree when a degree from Copenhagen
University would better prepare her for employment in the Danish school system. To what degree
Alfreds own precarious experiences in academic influenced his advice to Inge is unknown, but as a
women her job opportunities would be very limited and nearly non-existing at the university.

**4. Gap years**
Inge Lehmann took her father's concerns to heart and did not return to university. The next six
years of her life are sporadically illuminated in recently discovered autobiographical notes, written
much later in hindsight. In them, she acknowledged that acute overwork and a lengthy recovery
period led her to provisionally abandon her studies for the typical life of a middle-class working
woman.

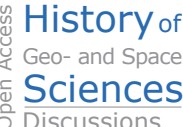

In the fall of 1912, a friend of her father's secured her an actuarial job at the insurance company *Det*
*Gjensidige Forsikringsselskab "Danmark"*. Her choice of employer was not unusual given that the
insurance business attracted many female academics with mathematical backgrounds. There, they
could use their statistical knowledge and calculating skills in office environments where female
clerks and typists had long been a common presence (Kragh, 2008).
The notes do not explain why Lehmann did not resume her studies as her father suggested. Possibly
her fatigue lingered longer than she had anticipated, or her family's financial needs were more
pressing. In any event, the outbreak of World War I in 1914 put an end to any thoughts of returning
to Cambridge.
Inge Lehmann remained at the insurance company for a number of years but expressed little interest
in the business aspects of her work (RA: Lehmann, biographical notes [u.d.]: W84-258079).[10]
When she was not promoted in step with her male colleagues, she recognized that gender was again
the restricting factor. Passed over for promotion and with the prospect of a male boss she found
unacceptable, she considered relocating to Canada, but another bout of overexertion prevented her
from emigrating.
Unable to secure a managerial position, Lehmann considered marriage. In February 1917, at the age
of 29, she became engaged and resigned from *Danmark*, as employment was incompatible with
matrimony. Only a month later she broke off the engagement in order to resume her studies and
pursue an academic career (RA: I. Lehmann, biographical notes [u.d.]: W84-2580).[11] Inge
Lehmann's decision to remain unmarried to further her academic ambitions was not an unusual
choice at the time. Abstaining from marriage was common for university women until the 1920s.
Thereafter, the number of married female academics increased but slowly (Rosenbeck, 2014). Inge
Lehmann embodied this trend as she remained unmarried, without children all her life.

**5. Return to the University of Copenhagen**
In August 1918, Inge Lehmann finally resumed her studies at the Faculty of Mathematical Sciences.
Two years later, she passed the second and final part of her examination with top grades, earning
her MA. It is worth noting that Lehmann's lengthy period of study manifested a general tendency
among female students at the Faculty. A survey of degrees completed between 1916-1920 at the
Faculty of Mathematical Sciences shows that a number of female students were enrolled for

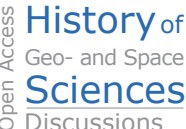

considerable lengths of time, and that female students in general were enrolled longer than their
male counterparts (Københavns Universitet, 1925).

Alfred Lehmann passed away in September of 1921. Among many other things, this meant that
Inge needed to secure a stable income. Also that year, an act was passed giving women equal access
to employment in the public sector, including all university positions. No longer forced to settle for
teaching, Inge Lehmann could now pursue a university career in mathematics with concomitant
salary, prestige and scholarly recognition.

### 5.1 Assistant in the Faculty of Mathematical Sciences

A small scholarship allowed Lehmann to study mathematics at the University of Hamburg for a
short period of time. After returning home again, she started work in March 1923 as assistant to
Professor Johan Frederik Steffensen in his Actuarial Mathematics Laboratory at the University of
Copenhagen. Inge's yearly income was DKK 700, plus a small bonus (RA: Københavns
Universitets Forsikringsmatematiske Laboratorium, Korrespondance: Konsostorium, letter 1. March
1923).[12] For this modest salary, she had to tutor students, assist in practicum sessions and grade
assignments. Grading mathematical problems after the practicums ate up a disproportionate amount
of her time, and it quickly became obvious that her income was not commensurate with the
demands of her duties.

Realizing this, Professor Steffensen tried on several occasions to secure better pay and conditions
for his assistant. In December 1924 he tried to get a reduction in her workload. A few months later
he complained to the Minister for Education that Lehmann's pay was considerably inferior to that
of other (presumably male) scientific assistants at the University and requested that it be brought up
to the same level as the others (RA: Københavns Universitets Forsikringsmatematiske
Laboratorium Korrespondance: Steffensen, letter 16. February 1925).[13] The gap between her salary
and that of the others must have been pitiful, because the Ministry of Education was quick to act: in
April her salary rose to almost three times its previous level (RA: Københavns Universitets
Forsikringsmatematiske Laboratorium, Korrespondance: Konsistorium, letter 30 September
1925)![14]

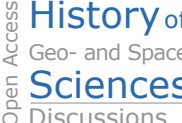

While working at the Laboratory of Actuarial Mathematics, Inge Lehmann had taken on part time jobs, including translation and editing for another Mathematics Faculty member: Professor Niels Erik Nørlund. In addition to his professorship, Nørlund had been appointed Director of the Danish Geodetic Service (Den Danske Gradmåling) in 1923, with a mandate to reform and merge the Service with the Topographic Division of the General Staff (Generalstabens Topografiske Afdeling).

The role of teaching assistant and occasional secretary was traditionally the end of the line for many women in academia, but Lehmann was not content in this final station. Having worked as Niels Erik Nørlund's occasional secretary, in June 1925 she cautiously pointed out to him that she wanted a research job: "I believe that I would venture to undertake calculation work, if it does not involve too great a theoretical foundation in areas with which I am not familiar, whereas I am not so certain that you would be served by my assistance with correspondence, as I understood to be your plan." (RA: N.E. Nørlund, letter (I. Lehmann) 17. June 1925)[15]

Nørlund could not employ her as research assistant at the university, but he saw another opening for her talent. He was in the process of reorganizing the Geodetic Service and needed to add seismological stations to their activities. An annual contribution from the Carlsberg Foundation made the project feasible, and for the next couple of years Inge Lehmann helped to set up the new seismological stations. In 1926 she helped establish seismic stations in Copenhagen (COP) and Ivittuut (IVI), Southwest Greenland, and in 1927 at Scoresbysund/ Ittoqqortoormiit (SCO), West Greenland (for the early history of seismology in Denmark, see Lehmann 1987, Jacobsen, 2017 , Dahl-Jensen, Jacobsen, Sølund, Larsen and Voss (submitted)).

Lehmann carried out the work of setting up and running the seismological stations in addition to her work at the Laboratory of Actuarial Mathematics. In January 1927, restructuring the Geodetic Service was so far advanced that she could resign from the Actuarial Laboratory and work exclusively for Niels Erik Nørlund. The plan was for Inge to learn the science of seismology so she could work in that field in the future.

As seismology in Denmark was in its infancy, Nørlund arranged for Lehmann to spend four months abroad in the autumn of 1927 to immerse herself in the science. Part of her time was spent at the

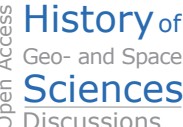

precursor of the International Association of Seismology and Physics of the Earth's Interior
(IASPEI), then known as the International Seismology Association of the International Union of
Geodesy and Geophysics (IUGG) (for the history of IASPEI, see Rothé, 1981; Schweitzer and Lay,
2019). The IUGG bureau was located in Strasbourg, there, she spent several weeks learning to read
seismograms. After attending the IUGG General Assembly in Prague, she put this skill to good use
while studying with Beno Gutenberg at his home in Darmstadt, Czechoslovakia (Lehmann 1987).

377       **6.  Director of the Seismology Department at the Danish Geodetic Institute**
In April 1928, Niels Erik Nørlund was appointed director of the newly formed Danish Geodetic
Institute (Geodætisk Institut). In May, Inge Lehmann was the second person in the country to sit for
the 'magisterkonferens' (equivalent to an MSc) in geodesy at the University of Copenhagen, a new
subject recently introduced at Nørlund's behest.

Her short apprenticeship abroad and her own studies were her only preparation for the examination,
which was tailored to her future job. In the written exam, she gave an 'Account of the key methods
for the determination of the epicenter of a seismic activity' (*Redegørelse for de vigtigste Metoder til*
*Bestemmelse af Epicentret for en seismisk Bevægelse*). Her final lecture considered cartographic
projection methods (Københavns Universitet, 1929), another essential area in the work of the
Danish Geodetic Institute.

By summer, Inge Lehmann was Director of the new Seismology Department at the Geodetic
Institute. She was responsible for running Denmark's seismological stations, along with a couple of
technical assistants. Although the job was mainly administrative and involved very little research, it
was a permanent position with the title and salary of a department head.

*Figure 3: Inge Lehman, Director of the Seismological Department of the Geodetic Institute, 1932*
*(Royal Danish Library)*



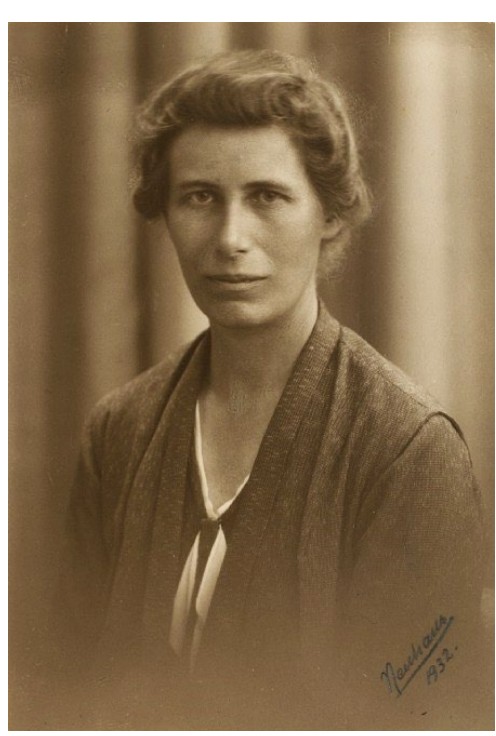



In a letter to Niels Erik Nørlund written that year, she expressed her pleasure and gratitude:
"I do not think I thanked you properly for my appointment […] I could not have wished for
anything better. I have earlier been concerned that I was asking too much when refusing to be
satisfied with working in order to earn money, but sought a job in which I could really take an
interest. In my work here, I have […] found more than I could ever have hoped. In return, I shall do
my utmost. It is no small thing to have the opportunity and permission to use all one's strengths."
(RA: N.E. Nørlund, letter (I. Lehmann) November 1928)[16]
Until she retired in 1953, Inge Lehmann was the only academic working at the Department of
Seismology. Due to her administrative duties, most of her research was performed in her spare time.
Overseeing stations in Denmark and Greenland gave her access to seismograms from several
locations and a range of instruments. As department head, she kept in contact with an international
network of colleagues. Her expertise in reading seismograms and vigorous correspondence with
leading seismologists paved the way for her discovery of the Earth's inner core in 1936, which
earned her lasting international renown as one of the most influential seismologists of the 20th
century (Hjortenberg 2009).

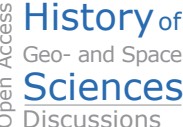


## 7. Discussion

As an early female scientist in Denmark, Inge Lehmann is virtually unsurpassed in the level of
employment she achieved and in the scientific recognition she received. However, her graduate and
postgraduate experiences reflect common features shared by most female academics of the time.
In her study of Danish female academics from c.1875 to c.1925, Rosenbeck (2014) identified four
commonalities. These women mostly came from affluent families or academic families. Female
students had higher average grades than their male counterparts, even though this gendered
difference diminished as the number of female students increased around 1900. Female students
also started their coursework far later than male students, although average age difference also fell
over subsequent generations. Finally, the vast majority of women academics remained unmarried.
Inge Lehmann's background and experience precisely fit in Rosenbeck's (2014) generalization of
female academics of the period: she came from an intellectual family, her grades were above
average, she took longer to finish her studies than the male students, and she remained unmarried.

Despite the fact that women were making their way in science by the 1920s, women academics did
not participate on equal terms with men. A number of societal and institutional factors in the natural
sciences contributed to women's continued difficulty in making a career (Kragh, 2008). The law
passed in 1921 giving women access to public sector employment was crucial for opening academic
appointments to college educated women in their research specialty – although in pay and prestige,
women still lagged behind men. As a rule, women were employed in positions where there was high
turnover in male personnel, or in newly established jobs.

American historian of science, Margaret Rossiter, in her cardinal work *Women Scientists in*
*America* (1984), showed that university-educated women often struggled with unemployment. Once
employed, their prospects for promotion were considerably inferior to those of their male
colleagues. In connection with female academics' career strategies around 1920, Rossiter pointed
out that many women turned to the "Madame Curie strategy": instead of addressing imbedded
inequality in the workplace, women often internalized their struggle. Wanting to prove their
entitlement to science, they tried to surpasse their male colleagues' scientific achievements. As a
result, some women drove themselves to exhaustion or nervous breakdowns in their quest for
academic excellence.  In the private industrial sector, women scientists were few and far between.

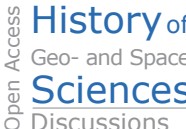

There, a second strategy of cynical versatility and conformity developed in the 1930s. Taking advantage of prevailing stereotypes, women deliberately sought jobs considered more adaptable to their gender, but close in proximity to their academic disciplines.

Margaret Rossiter's studies were based on the conditions of women in the US, but many of the patterns she observed can reasonably be applied to the situations of Danish female academics. From the first evident in 1911, Inge Lehmann displayed repeated, stressed-related breakdowns due to overexertion, a pattern of behavior analogous to Margaret Rossiter's observations about women's self-inflicted overcompensation.

Lehmann's appointment as Director of the Department of Seismology can also be interpreted from a gendered perspective similar to the cynical versatility Rossiter observed among female scientists in US industry. Niels Erik Nørlund's selection of Lehmann to manage the seismological stations was likely due to several factors in addition to her scientific qualifications. Firstly, there was no tradition of seismological research in Denmark, so this particular research area was not prestigious. Secondly, due to seismology's obscure status, there were no male candidates. Career prospects were limited in a country where earthquakes are extremely rare. Thirdly, the new job's responsibilities were mainly administrative and the Department's research was not connected to University of Copenhagen.
Nevertheless, some of the above mechanisms worked in Inge Lehmann's favor. By switching from mathematics to seismology and accepting a job outside the University, she secured a permanent appointment and realized her ambition of holding a senior scientific post at a time where faculty positions for women were extremely rare.

## 8. Conclusion

Among seismologists, Inge Lehmann is remembered for her uncompromising, sometimes undiplomatic ways (Bolt and Hjortenberg, 1994). But as a young woman, she was ambitious and adventurous, eager to experience life beyond Denmark. In becoming a scientist, her path was not straight forward. As a female she had to overcome society's general belief that women were biologically unsuited to academic studies, let along a scientific careers. She even had to overcome her father's belief that, while intellectual gifted, she was mentally and physically unfit for academic studies.

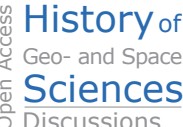

During her stay at Cambridge University in 1911, she had her first direct experience of gender-
based restrictions, and her mental breakdown in the winter of 1912 can be construed as an attempt
to rectify gender bias via academic overcompensation – a self-inflicted regimen, it must be said,
that Inge Lehmann shared with many contemporary female scholars.
In her work as an actuary and in her research assistant job, Inge Lehmann found herself in a
disagreeably inferior position compared to her male colleagues. When she changed her research
field from mathematics to the less prestigious seismology, she displayed a pragmatism that found
hope in what was possible and made the best of performing within a variety of narrow parameters
(only conducting research in her spare time, for example) in order to move up the career ladder.
Inge Lehmann had a career in science because at decisive moments she conformed to social,
professional and political agendas – and because she was a talented scientist.

**Disclaimer**
This paper is a revised version, with new data, of Jacobsen (2015).

**Acknowledgements**
I am indebted to author Lotte Kaa Andersen (lotte@kaaandersen.dk) for sharing her findings with
me and to independent researcher & editor Karen Alexander (piscepuella@gmail.com) for help
putting the manuscript together.

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
