# Peer review of "Title page: A seismologist's beginnings: Inge Lehmann's experiences during the 1910s and '20s as a woman in science."

_History of Geo- and Space Sciences, 2021_

## Author Response (AR1)

.

**Reply to reviewers – HGSSD 25. January 2022**

*Q2: Does the paper present new historic research, new interpretations or new compilations*

RC2: Yes, although it remains unclear what new results are presented in contrast to Jacobsen (2015). However, since Jacobsen (2015) was published in Danish, this article in English can reach a broader and more international audience.

> Reply: The introduction now specifies what new data are included and how it affects the conclusion compared to Jacobsen (2015).

*Q4: Do the authors give proper credit to related and previous work and clearly indicate their own new/original contribution?*

RC1: This could be clearer, what new archival work was undertaken to make this revision?

RC2: Yes. I would recommend to make a clearer statement about which new sources are presented and in what way the argument is original.

> Reply: Agreed. See reply to Q2.

*Q5: Does the title clearly reflect the contents of the paper?*

RC1:  It could be more descriptive and exciting.

> Reply: While I believe that the title accurately describes the content of the paper, I agree that it is not very exciting and will try to come up with something better.

*Q6: Does the abstract provide a concise and complete summary?*

RC1: The abstract is rather mundane and repeats materials from Lehmann's Royal Society biographical memoir, it should be more exiting as well.

RC2: The abstract does not contain the research question. It would be helpful to include the research question (or argument) and the method/sources that are used to answer it.

> Reply: The reviewers have an excellent point and I have amended the abstract accordingly.

*Q9: Should any parts of the paper (text, formulae, figures, tables) be clarified, reduced, combined, or eliminated?*

RC1: NO, but I recommend the author examine the experience of women in science in other nations (beyond Denmark and America) , for example in other Nordic nations, UK, and possibly Germany.  For

example, here is list of women who received advance training in meteorology before 1930 under N. Shaw and V. Bjerknes

- Marie Dietsch, D. Phil. mathematical geophysics, 1918, Leipzig (V. Bjerknes)
- Elen Elaine Austin, BSc. maths and natural sciences, 1918, Cambridge, (N. Shaw)
- Luise Charlotte Lammert, PhD climatology, 1919, Leipzig (V. Bjerknes)
- Anne Louise Beck, MS geography, 1922, Berkeley (V. Bjerknes)

I like the table showing Danish laws for women's rights, but I also think a short comparative study of women in science would be helpful.

> Reply: The response raise an interesting conundrum well known in science studies. If we cannot make general assumptions based on the extraordinary, why study such gifted individuals at all? What is the purpose of biographies of scientists? Quoting Helge Kragh: "On scientific biography and biographies of scientists" in *Relocating the History of Science* (2015) "One of the advantages of the biographical method is that it stimulates a more integrated and coherent picture of science, if limited to a unique case only, precisely of its focus on the individual scientist."
>
> By putting Inge Lehmann into a social context, we discover structures in science that might be overlooked in more generalized studies. In Inge Lehmann's case it is also very difficult to do a comparative study because of her unique status. The problem is not solved by extending the study to female scientists in other nations because of their different cultural backgrounds. As my article documents the experiences for women studying in Cambridge was vastly different from studying in Copenhagen which again was different from studying in Germany or USA, making transnational comparisons highly speculative.
>
> However, I acknowledge the benefit of providing some additional contexts to Inge Lehmann's experiences and have provided some examples of contemporary female academics educated or working in Denmark.

Also, the question of perceptions of Inge's fortitude and ability to conduct scientific campaigns in the field, especially in the Arctic which was traditionally seen as a man's domain, could be further explored.

> Reply: Artic expeditions and station's life in the Artic was very much a man´s domain (and possibly still is). Point in fact paleo-climatologist Willi Dansgaard infamously excluded female scientists from his expeditions until the late 1970s.
>
> Inge Lehmann never worked in the Arctic, and she, like most other seismologists at the time of her research, relied on seismograms sent to her. The only time we know she was in Greenland was two weeks around 1934, when she supervised the installation of instruments at SCO after a fire. As a mountaineer, she was physically capable, but as the head of department it was inconvenient for her to be out of reach for long periods of time.

RC2: I suggest some more clarification regarding the following issues:

In order to contextualize, I would recommend to include some more information about how Lehmann's case is not only representative for other women but also how representative it was for Denmark/Danish femal scholars (within and beyond the university, and in a more contextualized form than the otherwise still helpful table 1). This would be helpful for readers who are not familiar with Danish social history.

Reply: See above reply

The originality of the paper could be made stronger by clarifying the questions: Why is this new material really important? Does it tell a new story or does it make a known story more nuanced? The result (Lehmann's career was typical for a female academic) is not surprising and doesn't seem very original at first sight. But maybe Lehmann's case is indeed more interesting than other cases of female academics in other scientific disciplines? The argument that Lehmann succeeded only because seismology was new and less prestigious sounds plausible and really interesting. As a suggestion, this argument could be stronger, more prominent and more elaborated in order to emphasize the relevance and uniqueness of the case.

Reply: I have rewritten the introduction and added to the discussion to show why Inge Lehmann's story is relevant to history of science and to the discussion of women in STEM. See also my reply to Q9

Also, including some Gender Studies literature (in addition to Rossiter) and referring to the current topics in Gender Studies might help to justify the relevance of Lehmann's case beyond "just adding another story" of a woman in science. But also without this specific addition, I think it is an interesting paper worth to be published with minor revisions.

Reply: Given the profile of HGSS I think a larger discussion of Gender Studies literature and modern perspectives on women in STEM is better published elsewhere.  But the reviewer's point is well taken, and it is a topic I plan to explore further in another publication.